# IoT for Smart Cities: Machine Learning Approaches in Smart Healthcare—A Review

**Taher M. Ghazal [1,2]**, **Mohammad Kamrul Hasan [1]**, **Muhammad Turki Alshurideh [3,4]**, **Haitham M. Alzoubi [5,*]**, **Munir Ahmad [6]**, **Syed Shehryar Akbar [6]**, **Barween Al Kurdi [7]** and **Iman A. Akour [8]**

1    Center for Cyber Security, Faculty of Information Science and Technology, Universiti Kebangsaan Malaysia (UKM), Bangi 43600, Selangor, Malaysia; ghazal1000@gmail.com (T.M.G.); mkhasan@ukm.edu.my (M.K.H.)
2    School of Information Technology, Skyline University College, Sharjah 1797, United Arab Emirates
3    Department of Marketing, School of Business, The University of Jordan, Amman 11942, Jordan; malshurideh@sharjah.ac.ae
4    Department of Management, College of Business, University of Sharjah, Sharjah 27272, United Arab Emirates
5    School of Business, Skyline University College, Sharjah 1797, United Arab Emirates
6    School of Computer Science, National College of Business Administration & Economics, Lahore 54000, Pakistan; munir@ncbae.edu.pk (M.A.); syedshehryar@live.com (S.S.A.)
7    Department of Business Administration, Faculty of Economics and Administrative Sciences, The Hashemite University, Zarqa 13115, Jordan; barween@hu.edu.jo
8    Department of Information Systems, College of Computing and Informatics, University of Sharjah, Sharjah 27272, United Arab Emirates; iakour@sharjah.ac.ae
*    Correspondence: haitham.alzubi@skylineuniversity.ac.ae

**Abstract:** Smart city is a collective term for technologies and concepts that are directed toward making cities efficient, technologically more advanced, greener and more socially inclusive. These concepts include technical, economic and social innovations. This term has been tossed around by various actors in politics, business, administration and urban planning since the 2000s to establish tech-based changes and innovations in urban areas. The idea of the smart city is used in conjunction with the utilization of digital technologies and at the same time represents a reaction to the economic, social and political challenges that post-industrial societies are confronted with at the start of the new millennium. The key focus is on dealing with challenges faced by urban society, such as environmental pollution, demographic change, population growth, healthcare, the financial crisis or scarcity of resources. In a broader sense, the term also includes non-technical innovations that make urban life more sustainable. So far, the idea of using IoT-based sensor networks for healthcare applications is a promising one with the potential of minimizing inefficiencies in the existing infrastructure. A machine learning approach is key to successful implementation of the IoT-powered wireless sensor networks for this purpose since there is large amount of data to be handled intelligently. Throughout this paper, it will be discussed in detail how AI-powered IoT and WSNs are applied in the healthcare sector. This research will be a baseline study for understanding the role of the IoT in smart cities, in particular in the healthcare sector, for future research works.

**Keywords:** smart cities; IoT; machine learning; sensor networks; artificial intelligence; healthcare



## 1. Introduction

There is hardly an industry that does not benefit from increasing IT resources and self-learning computer networks—but in the healthcare sector, AI can be used in different ways, because AI approaches help with treatment and especially with operations, but also support medical staff in diagnostics and prophylaxis. In the USA, for example, there are already operating rooms in which all steps of an operation are meticulously recorded and the treating physicians are also given detailed support in deciding on the next steps and even the individual incisions to be made. The patient is tracked at all times by sensors built into the room and thus optimally cared for.

The work of specialists, for example, that of a radiologist, will also change as a result. Especially with imaging processes that can be easily evaluated and standardized technically, the technology is already doing a remarkable job. Since it has evaluated millions of cases, it can judge with greater accuracy than the treating doctor with the naked eye. Even if the decision-making authority will always remain with the doctor, thanks to artificial intelligence he will receive a valuable second opinion that can underpin his own anamnesis or add further aspects.

The more data the system has from the patient himself and the more other cases it knows, the more certain the statements and recommendations for action will be. It will therefore be desirable in the future to have as much data as possible available and to evaluate it, considering all anonymization measures. In general, data from wearables in everyday diagnostics will support the patient and the treating doctor in making the right decisions more quickly and making recommendations for action. For example, a patient who wakes up in the morning with an increased pulse and blood pressure can better decide whether to see a doctor or whether the malaise is likely to go away on its own in the course of the morning.

The challenge we face today is that we are struggling with various interfaces between medical subsystems that have to ensure interoperability with one another. With the help of a blockchain, patient data records can be standardized, reliably exchanged, evaluated and stored in a forgery-proof manner. This means that those responsible for medical data do not have to harmonize several different systems in parallel, but can rely on a reliable standard [1]. A blockchain as a basic technology will also be used in other industries and sectors where the exchange of reliable, unchangeable data is important. We would be an important step closer to the digital patient file, which provides the treating doctor with all relevant data (and thus also benefits the patient). In a broader perspective we can refer to blockchain technology in the IoT as a "secure by design" system that can be used to address security concerns in IoT applications, considering the impressive features of blockchain technology such as immutability, transparency, auditability, data encryption and operational resilience, as explained by [2].

Networked smart home applications will also be helpful in this context. The patient, who gets up in the morning and stands in front of a mirror equipped with cameras, will be able to establish a connection with the attending physician if he has symptoms. Based on the patient data, he then decides in real time which steps are sensible. However, the whole thing goes even further: in the worst case, fall sensors installed in the carpet and in the house or suitable healthcare wearables could detect when a patient falls and is lying on the floor or when help is required for other reasons. Automated in the context of a smart city, the corresponding control center could be informed and an ambulance sent, with the appropriate hospital with the appropriate capacity being selected based on the diagnosis. In combination with smart diagnostics and wearables, such smart infrastructures could help older patients in particular to live independently longer than before in their own homes and still not have to do without reliable mechanisms for care.

The acquisition of patient data takes up a (too) large part of the limited time of nursing staff, especially in the hospital environment. Here, intelligent devices can be used to collect the important patient data via voice command, for example, and assign it correctly and evaluate it with the help of artificial intelligence and big data. This cannot all be recorded using wearables but is often also obtained from the impression and assessment of the patient by the nursing staff. The nursing staff would then have more time for the actual care and would not have to record all the data or even later transfer it to the electronic patient file via a PC. Basically, this could be a (partial) answer to the acute emergency care needed in the care sector, because if we look at the shortage of skilled workers and personnel, it quickly becomes clear that this is one of the greatest challenges of the future in view of an increasingly aging society [1]. Figure 1 shows a substantial proportion of the AI literature analyses data from diagnosis imaging, genetic testing and electrodiagnosis.

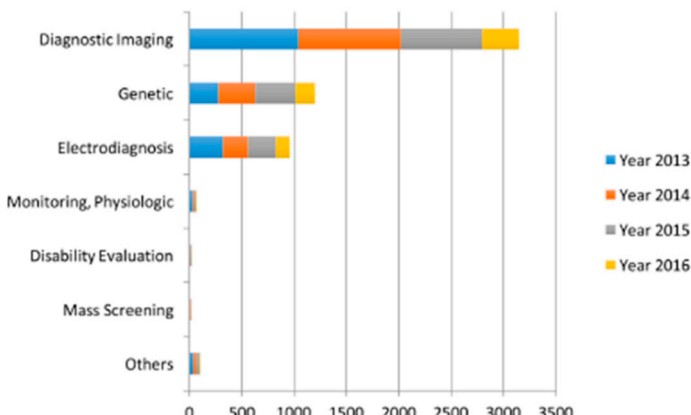

**Figure 1.** Types of data considered in AI related studies (PubMed database) [3].

A medical app could not only issue a warning if the doctor wanted to prescribe a drug for which an intolerance is noted in the medical record, it could also indicate that the patient has a genetic trait that makes the drug less effective. AIs already obtain this information from studying numerous specialist publications or by searching through databases with countless treatment histories of other patients. If the program detects certain patterns— same genetic trait, same medication, same results—it recommends an alternative healing approach.

It is also conceivable that the AI will make predictions and suggest appropriate drugs that could be important for a patient in the future. It could also provide brief summaries of the medical history and, depending on the symptoms, highlight information that is hidden in the depths of the patient files. Socio-economic and demographic data can also be considered. Neural networks are already helping with the early detection of breast cancer [4].

If it were possible to implement a comprehensive, data-sensitive digitization of the healthcare sector, the hospital staff would be relieved considerably. In addition to the more effective diagnostic and documentation methods described above, there would also be the option of short-circuiting AIs with IoT networks. With centralized device monitoring, biodata could be transmitted in real time and analyzed by an AI. Basic functions of hospital organization, such as monitoring material consumption, could also be handed over to computers. It would also be conceivable to expand decentralized medical care for immobile patients or even for those who are thousands of kilometers away. Where there is a lack of specialists and modern analysis tools, it may soon be enough to send a photo or a set of blood values online to a computer center and wait for the diagnosis. Lives could be saved in this way, especially in medically undersupplied regions.

There is another important aspect related to smart cities, rather a critical view, since there are a lot of benefits, there are potential downsides as well for the technology; over the past few years, urban experimentation has become a popular norm where alternative models of urbanization are being developed to address issues of sustainability in cities. Cugurullo [5] counterclaims general understandings of smart and ecological urbanism, arguing that what are promoted as cohesive settlements shaped by a homogeneous vision of the sustainable city, are actually fragmented cities made of disconnected and often incongruous pieces of urban fabric. Cugurullo [5] identified five forms of vulnerabilities with respect to smart city technologies and also explained the present degree of cyberattacks on network infrastructure and services. Cugurullo's [6] work on understanding urban studies and planning is of great impact. It explains in great detail the purpose of cities in a time where human and artificial intelligence are irreversibly colliding in the built environment.

Following are the motivational factors for this literature review:

- Perform a baseline study focusing on:

- IoT
- Smart HealthCare
- Smart Cities
- AI/ML in Smart Cities
- Blockchain in Smart Cities
- Study existing literature;
- Study wireless sensor networks in relation to the IoT and smart cities;
- Understand and relate the IoT with smart healthcare;
- Study AI/ML applications in smart healthcare.

This paper aims to provide a baseline study toward IoT, smart healthcare, smart cities, machine learning and their co-relation. It has been organized into six sections. Section 1 contains a brief and comprehensive introduction. In Section 2, related work is discussed. In Section 3, wireless sensor networks (WSN) and their contribution in IoT-based smart cities and healthcare is explained. In Section 4, the relationship between the IoT and health care systems is discussed with brief descriptions of IoT applications in different health care systems and scenarios. Section 5 explains role of machine learning in conjunction with the IoT and smart healthcare systems, and finally, in Section 6, conclusions are drawn.

## 2. Related Work

In this section literature related to the IoT, smart cities, smart healthcare, blockchain and AI in the IoT is reviewed. Habibzadeh et al. [7] studied emerging trends in smart healthcare applications and key technological developments that have a direct impact in these transitions. The authors also reviewed different security considerations in smart healthcare systems and their consequences and counter measures. Latif et al. [8] analyzed security and privacy issues in the IoT's application in smart cities and identified their proposed solutions. Authors tend to use graph theory for rectification of highlighted issues since the conventional approaches do not provide optimal results for security and safety critical systems. The authors in [9] consider smart city systems as a whole and discuss how a smart healthcare system interacts and collaborates with smart city infrastructure and how it works for the healthcare field. Several case studies are also reviewed by the authors and they suggest how a more powerful, integrated and effective system can be built out of smart healthcare systems. A comprehensive survey study was performed by Arasteh et al. [10], describing IoT technologies for smart cities and the main components and features of a smart city. The authors also explained practical experiences and challenges faced by implementers around the world. Alsamhi et al. [11] presented a comprehensive survey of potential techniques and applications of collaborative drones with the IoT which have been recently used to enhance the smartness of smart cities based on data collection, privacy, security, public safety, disaster management, energy consumption and quality of life in smart cities. Li et al. [12] presented a comprehensive review of the applications of machine learning techniques for big data analysis in smart healthcare systems. The authors also highlighted several strengths and weaknesses for existing approaches with a special focus on research challenges in this aspect. Kharel et al. [13] proposed a model for a smart health monitoring system based on fog computing. The proposed architecture claims to acknowledge the underlying problems of a clinic-centric healthcare system and change it to a smart patient-centric healthcare system. Muhammad et al. [14] presented a comprehensive review of the published surveys using deep learning-based methods for brain tumor classification, covering the main steps including preprocessing, features extraction, classifications, achievements and limitations. The authors also investigated state-of-the-art convolutional neural network models for BTC by performing extensive experiments using transfer learning with and without data augmentation. Soomro et al. [15] proposed a model combining artificial intelligence with the IoT to reduce traffic congestion in a smart city environment. Allam et al. [16] reviewed the urban potential of AI and proposed a new framework binding AI technology and cities, ensuring the integration of key dimensions of culture, metabolism and governance, often considered key factors in

the successful integration of smart cities. Majeed et al. [17] presented a comprehensive review of the evolution and role of blockchain in the development of IoT-based smart cities. Blockchain evolution in terms of constituent technologies, consensus algorithms and blockchain platforms are presented in step 1. In the second step, several smart applications enabled by blockchain are discussed and evaluated, and finally, real world blockchain applications in smart cities are discussed as case studies. Cugurullo [18] explored the theory and practice of artificial intelligence's intersection with the development of smart cities in three steps. First of all, the authors present a theoretical framework for better understanding of AI in the context of urbanism. Secondly, they examined the case study of Masdar city, which is an Emirati urban experiment, and thirdly, they proposed a research agenda for the investigation of an autonomous city. Ullah et al. [19] performed a review to evaluate the roles of artificial intelligence, machine learning and deep reinforcement learning in the evolution of smart cities. Jha et al. [20] performed a study to develop an effective system for possible monitoring of losses, traffic management, smart city innovations, digitalized and integrated systems and software. Singh et al. [21] introduced an ML-based, distributed big data analysis framework for next gen web for the IoT and demonstrated that the suggested distributed framework is more efficient than the conventional frameworks. Singh et al. [22] proposed DeepBlockScheme, which is a deep learning-based blockchain scheme for secure smart cities. The authors presented a case study of car manufacturing for the proposed scheme and compared it with existing research with parameters such as security and privacy. Energy efficiency in the IoT is also one of the key areas of research. Zhang et al. [23] systematically analyzed IoT architecture and the power distribution within. A comprehensive summary of the energy consumption model in the IoT and pros and cons of improving energy efficiency were presented. Singh et al. [24] proposed an OTS (one-time signature) scheme-based secure architecture for an energy-efficient IoT in edge infrastructure. The authors used a blockchain-based, distributed network at the fog layer to improve the security and privacy of the architecture.

A lot of research has been carried out over the course of the past few years for the IoT, smart cities, smart healthcare, blockchain and AI in the IoT. Use of WSN in smart cities and in particular healthcare has opened up several new research avenues and challenges for smart cities which are addressed by employing AI and blockchain technologies. The following Table 1 presents a comprehensive view of the reviewed literature for this study.

**Table 1.** Reviewed literature.

| Sr. | Title | Covered Areas | | | |
| --- | --- | --- | --- | --- | --- |
| | | **Smart Cities** | **Smart Healthcare** | **AI/ML** | **Security** |
| 1 | Toward uniform smart healthcare ecosystems: A survey on prospects, security, and privacy considerations. [7] | N | Y | N | Y |
| 2 | A survey of security and privacy issues in IoT for smart cities [8] | Y | N | N | Y |
| 3 | A Survey on the Status of Smart Healthcare from the Universal Village Perspective [9] | Y | Y | N | N |
| 4 | IoT-based smart cities: A survey [10] | Y | N | N | N |
| 5 | A survey on collaborative smart drones and internet of things for improving smartness of smart cities. [11] | Y | N | N | N |
| 6 | A Comprehensive Survey on Machine Learning-Based Big Data Analytics for IoT-Enabled Smart Healthcare System [12] | Y | Y | Y | N |

**Table 1.** *Cont.*

| Sr. | Title | Covered Areas | | | |
|---|---|---|---|---|---|
| | | **Smart Cities** | **Smart Healthcare** | **AI/ML** | **Security** |
| 7 | An architecture for smart health monitoring system based on fog computing [13] | N | Y | Y | N |
| 8 | Deep Learning for Multigrade Brain Tumor Classification in Smart Healthcare Systems: A Prospective Survey [14] | N | Y | Y | N |
| 9 | Artificial Intelligence enabled IoT: Traffic congestion reduction in smart cities. [15] | Y | N | Y | N |
| 10 | On big data, artificial intelligence and smart cities. [16] | Y | N | Y | N |
| 11 | Blockchain for IoT-based smart cities: Recent advances, requirements, and future challenges. [17] | Y | N | N | Y |
| 12 | Urban artificial intelligence: From automation to autonomy in the smart city. [18] | Y | N | Y | N |
| 13 | Applications of artificial intelligence and machine learning in smart cities [19] | Y | N | Y | N |
| 14 | Mitigating and monitoring smart city using internet of things [20] | Y | N | N | N |
| 15 | Machine learning based distributed big data analysis framework for next generation web in IoT [21] | Y | N | Y | N |
| 16 | DeepBlockScheme: A Deep Learning-Based Blockchain Driven Scheme for Secure Smart City. [22] | Y | N | Y | Y |
| 17 | Energy efficiency in internet of things: An overview [23] | N | N | N | Y |
| 18 | OTS Scheme Based Secure Architecture for Energy-Efficient IoT in Edge Infrastructure [24] | Y | Y | N | Y |

## 3. Wireless Sensor Networks for Smart Cities

A wireless sensor network (WSN) is a computer network comprised of sensor nodes which are organized in one network to monitor their surroundings using sensors Figure 2. While the first sensor networks consisted of comparatively few, largely wired and relatively large sensors, this changed at the end of the 1990s. In particular, the progressive miniaturization of processors, radio modules and sensors finally led to the introduction of wireless sensor networks at the turn of the millennium. These are networks without a fixed infrastructure between the sensor nodes themselves and the end devices (gateway). The origin of wireless sensor networks lies in the field of wireless, self-configuring ad hoc networks. Ad hoc networks are meshed networks in which the sensor nodes and end devices spontaneously connect to one or more of their neighboring nodes without defined hierarchies and can thus exchange data or commands; thereby, multi-hop communication is also enabled, in which data is passed from node to node until it reaches its destination. However, this also means that these networks are characterized by unpredictable, dynamic behavior. The network topology is different from permanently installed computer networks, as there are no fixed specifications for the network infrastructure. Thus, nodes can be added during operation or without warning. The strengths of ad-hoc networks with multi-hop are that they have a much simpler configuration, are more stable communication and have self-healing capabilities [25].

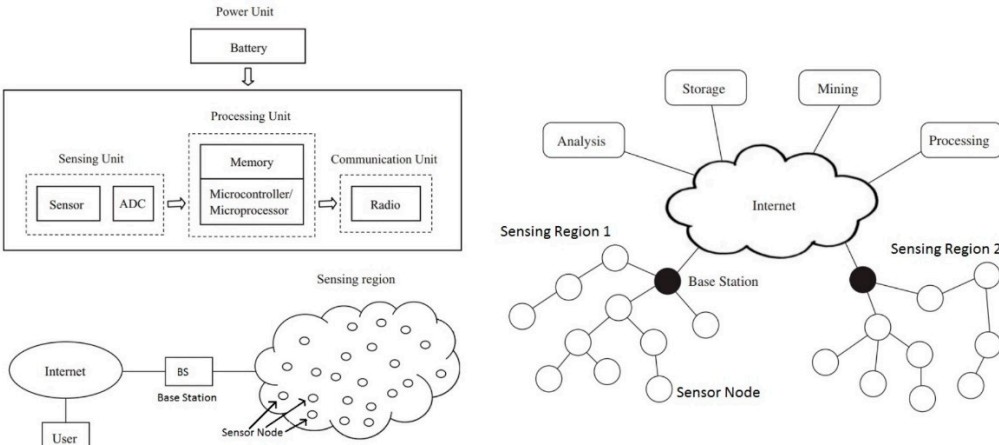

**Figure 2.** Graphical representation of WSNs for smart cities [26].

Through the interaction of individual or many sensor nodes that communicate via radio, large network structures can be set up that allow extensive observation and monitoring of various phenomena. Measurements can be carried out at high frequency and with great accuracy without disturbing or influencing the processes to be observed. All the sensor data are forwarded to the base station and the control of the network is organized via wireless communication. Both data types are cascaded or protocol-controlled from and to the control point, the gateway, transmitted through the network and forwarded from there to the control unit (query system/server). The main advantages of many modern wireless sensor networks are their ad-hoc and multi-hop capabilities, autonomous intelligent operation, ability to fuse sensors, bidirectional communication, good value for money, energy efficiency and self-healing capabilities [27]. Wireless sensor networks work transparently, i.e., they offer a platform for the transmission of any measurement data without knowing its meaning.

The sensing modules for wireless sensor networks have a strictly modular structure and provide various open interfaces for connecting digital and analog sensors. As a result, a wide variety of sensors can be integrated and addressed relatively flexibly. A wide variety of sensors are compatible, ranging from basic temperature and humidity sensors to high-precision vibration sensors or GPS modules for position determination. Thanks to improved manufacturing processes, standardization and advances in the field of microsensors (MST), very small, inexpensive, but nevertheless, very precise measuring sensors, so-called MEMS, can be integrated into such systems today.

The requirements for sensor networks are usually very different from those of conventional networks. These requirements must be met by special communication protocols, and algorithms are treated and simply cannot be taken over. Thus, all nodes need a high level of self-organization to build a functioning network structure. Here, nodes of the network fail and others join, without affecting communication, otherwise the network may collapse. Conventional communication principles such as client-server models are unsuitable for this. To be monitored by such sensors, event-based solutions are more suitable. Here, a message is sent by a sensor as soon as a certain limit value of the sensor is exceeded. To get the information about these events through the network to self-organizing routing nodes requires special cooperative algorithms. Sometimes, the collected information in the node is already preprocessed to announce information about the entire topology of the network. In addition to these dynamic requirements, sensor networks are subject to restrictions as the individual sensors only have limited energy resources, and these, if possible, are to be used sparingly. Thus, the knots should work effectively and, with a minimum, get by on control packages, e.g., parts of the node or the node itself is put into sleep mode if no activity in the monitored area is recorded and only wakes up when communication is necessary [28].

In addition to the problems that have to be considered when developing MAC protocols, questions of routing are also of great importance. Here, too, conventional technologies and algorithms cannot simply be adopted. The main problem researchers struggle with is the strong dynamics typically found in sensor networks. Accounts can permanently change their position and thus make the creation of stable routing paths impossible. All nodes have limited energy resources and can suddenly fail, and yet, the data traffic routed from accounts to nodes must not come to a standstill. Although the routing protocols have a number of difficult problems to solve, they still have to be relatively simple and easy to implement so that none of the nodes have to spend a lot of energy on routing [25]. Despite all of the changed conditions compared to conventional networks, tried-and-tested mechanisms have also been adopted. In this way, every node can fall back on a flood of its data in the event of an error. The packet in question is simply sent to all nodes within range with the hope of reaching the destination. Of course, this is by no means an optimal solution, as too much network traffic is generated and an enormous amount of energy is wasted. In addition to very simple processes such as shortest-hop-first, in which the packet is sent to the node that is closest to the destination, or hierarchical processes in which tree-like structures are built, one also tries to break new ground. Thus, probabilistic methods are often used to keep the routing effort in each individual node low. Similar to MAC protocols, it remains to be seen that no routing protocol will fully establish itself because the context of the application determines the most suitable protocol. In addition to the actual application, the characteristics of the network traffic are often analyzed and design decisions for the routing protocol are made on this basis [28].

One of the major challenges faced by WSNs is cheap production of small, robust sensor nodes for widespread deployment. There are an increasing number of small businesses operating that manufacture WSN hardware, and the market situation is similar to that of home computers in the 1970s. A great deal of R&D is being carried out to improve both the hardware and software aspect of the sensor nodes and networks. At the same time, there is also great focus towards making the WSNs more energy efficient considering both the node and network aspects of WSNs. In most modern examples, gateways are used to communicate between the WSNs and the LAN and WAN. The gateways bridge the space between the WSN and the other network. This enables the data to be stored and processed by devices remotely, e.g., on a remote server. The network used for such low-power wide area communication is known as a LPWAN (low-power wide area network) [27].

The networking and communication aspect of WSNs is regulated by several standards, and a range of application-specific solutions are available. Thread and ZigBee can connect sensors with 2.4 GHz and a data rate of 250 kbit/s. Many use a lower frequency to increase the radio range (typically 1 km), for example, the Z-wave works at 915 MHz and in the EU 868 MHz is widespread, but a major drawback is a significantly lower data rate (typically 50 kb/s). The IEEE 802.15.4 working group provides a standard for connectivity for low-power devices. Typically, sensors and smart meters use one of these standards for connectivity [6]. In the context of the IoT many other solutions are available. LORA is a form of LPWAN that provides long-range, low-power wireless connectivity for devices that have been used in smart meters and other long-range sensor applications. Wi-SUN connects devices at home. NarrowBand IOT and LTE-M can tether up to millions of sensors and devices using conventional cellular networks.

## 4. IOT and Healthcare

In conjunction with WSNs the IoT brings an unprecedented amount of data that the network infrastructure needs to handle. The solution for these problems is to customize the traditional network designs to the latest standards of network intelligence, which ensures optimal security [29]. Hospitals, clinics and care facilities need one cost-effective network infrastructure, the security of which complies with data protection regulations but is also easy to use and to operate. Figure 3 shows the leading disease types considered in the artificial intelligence literature.

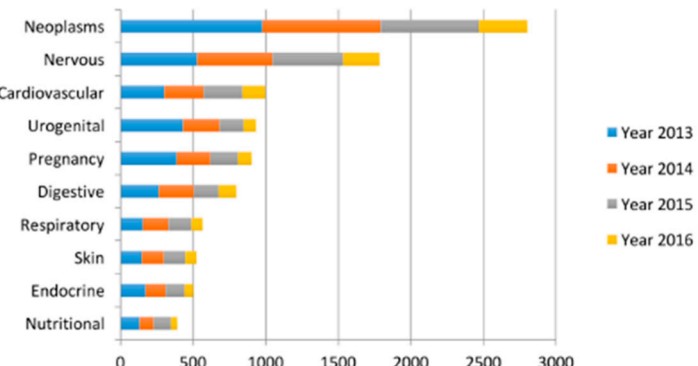

**Figure 3.** The leading disease types considered in the artificial intelligence literature (PubMed) [3].

### 4.1. Patient Monitoring

Advanced technologies in health care allow both inpatient and outpatient care to be monitored more closely. Remote patient monitoring (RPM) enables healthcare professionals to monitor vital signs and assess physical reactions to previous treatments without having to be in the same place as the patient. The device used depends on the health of the respective patient. For example, it could be an implanted cardiac device, an airflow monitor, or a networked blood glucose meter. The device in question collects the desired data. If the values are not as they should be, the data are simultaneously forwarded to a database for recording and to the treating doctor. The doctor can analyze the information in real time and react accordingly. Such devices are often used immediately after an operation. They help reduce the number of hospitals stays and avoid re-admissions because problems are identified more quickly. This allows doctors to react earlier and avoid potential complications. With the help of the data collected in real time, it is also possible to adjust and adapt treatment options more quickly, depending on the patient's physical reaction and condition. This allows doctors to react earlier and avoid potential complications [1].

### 4.2. Digital Drugs

One of the newer innovations in the healthcare industry is what is known as "smart pills". Smart pills are taken like normal medication but are equipped with some kind of monitoring technology in addition to the actual medication. They use it to forward information to a sensor worn on the body. These sensors monitor drug levels in the body based on a patient's perceived or diagnosed condition. The data from the portable sensors are then transmitted to a mobile phone app, which means that patients can access data on their vital functions themselves. Doctors can do this if the patient agrees. This is how the treating physicians determine whether a drug is working as intended or possibly causing side effects [1].

In November 2017, Abilify MyCite launched the first FDA-cleared smart pill that timestamped only when the drug was actually taken. As soon as the pill comes into contact with the patient's gastric acid, it triggers a sensor that marks the time of contact and first forwards this information to the wearable sensor and finally to the mobile phone app. The correct dosage and the prescribed intake are important prerequisites for a successful treatment. Such information is very valuable to medical professionals, and they no longer have to rely on the patient's word alone when treatment plans must be strictly adhered to. If patients fail to do this, the doctor can seek a discussion and clarify the cause directly [25].

One of the areas that causes a little more discomfort are so-called "robotic" pills. Once ingested, they are able to perform certain functions directly in the patient's body. Companies like Rani Therapeutics are developing pills that have the ability to pass through the body to navigate and perform functions that, for some reason, cannot be performed non-invasively. For example, Rani invented a pill that navigates through the stomach into the small intestine, where it injects without the exposure of the active compound to

digestive enzymes. When the drug is delivered, the residue dissolves and is digested, an option that is ideal for large, long, chained drug molecules such as proteins, peptides and antibodies.

### 4.3. Medical Equipment

Wearable medical devices are the most attractive option today for consumers of all ages for tracking their own vital signs in real time. In addition to Fitbit, Apple Watch and Co., other wearable technologies have now been developed. Not only do they record data, but they also perform certain functions based on commands or recognized situations. One example is "intelligent associations". They are equipped with sensors that assess the size of the underlying wound to determine whether it is healing or not, or if there is an infection, and whether a topical solution may need to be administered [25].

"Networked contact lenses" are also a form of wearables in health technology. In 2014, Google and Novartis began developing a connected contact lens that could monitor blood sugar levels by analyzing the patient's tear fluid. The data collected via the contact lenses is then sent to an insulin pump and the patient is informed whether their blood sugar level has reached a dangerous level and needs to be adjusted. This advance in non-invasive monitoring of diabetic patients could be life-changing for many. Because many patients suffer from having to prick themselves several times a day in order to measure their blood sugar level, such innovations give diabetes patients hope that non-invasive techniques are actively being sought and that they are on the verge of becoming a reality [30].

### 4.4. Medical Institutions

Many of the benefits of the IoT for the healthcare industry lie in improving the quality of care for patients. However, thanks to the Internet of Things, medical facilities have also improved, for example through more efficient processes and by conserving valuable resources. Intelligent technology in hospitals and care facilities ensures, for example, that doctors can better monitor expensive devices such as MRIs, CT and PT scanners and X-ray machines in terms of effectiveness and service life. In this way, malfunctions or incorrect operation can be avoided. Remote sensors minimize the number of manual tests or may even eliminate the need for them. That frees up time for more urgent tasks. A common problem in medical facilities is the relocation of equipment or systems that are used very often. This becomes a problem if a device cannot be located in an emergency. The use of Bluetooth low-energy location technology enables devices to be located in real time. This avoids anger and stress if you cannot find a device in an emergency situation. A small innovation helps to save countless lives. In contrast, the costs are negligible. Healthcare is one of the industries where the IoT has already made its mark [31]. A 2017 study by Aruba Networks found that 60 percent of healthcare organizations around the world have already implemented IoT devices in their facilities, and that number will grow dramatically in the years to come. It is obvious that IoT solutions have found their way into the healthcare industry and have become indispensable there. From better patient monitoring to intelligent pills to low energy location solutions, the IoT makes life easier for medical professionals and improves the treatment and care of patients, ultimately the goal of the IoT—to improve the quality of life for as many people as possible.

## 5. Machine Learning

Whether in prevention, early diagnosis of illness or the right choice of therapy, artificial intelligence and machine learning in combination with IoT-enabled WSNs can make a significant contribution in the healthcare scenario. In the future, better and more personalized medical care can be provided. Machine learning is a key technology in artificial intelligence. It requires a large amount of sample data, on the basis of which special algorithms develop models using pattern recognition. In the next step, these models can be upgraded and applied to unknown situations [32]. Figure 4 shows general classification of machine learning algorithms.

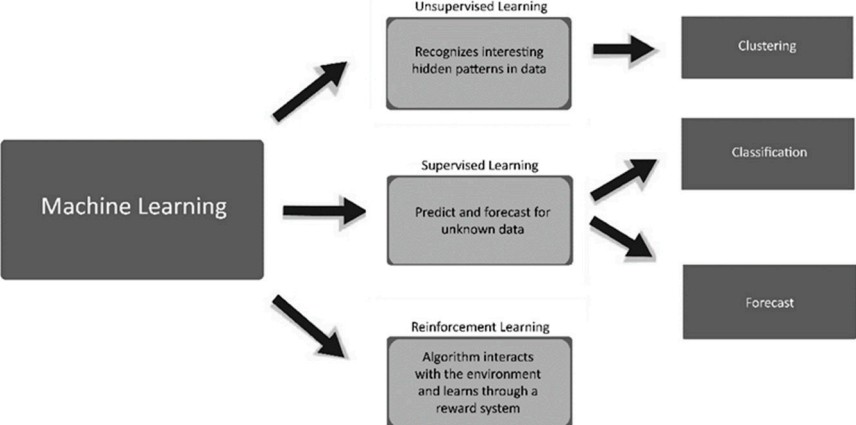

**Figure 4.** General classification of machine learning algorithms.

A basic distinction is made between the learning styles as supervised, unsupervised and reinforcement learning. Different procedures and methods are used in all three learning styles. Recently, deep learning has become more and more important in this context since it can recognize complex patterns of big data. Figure 5 shows machine learning algorithms used in healthcare sector.

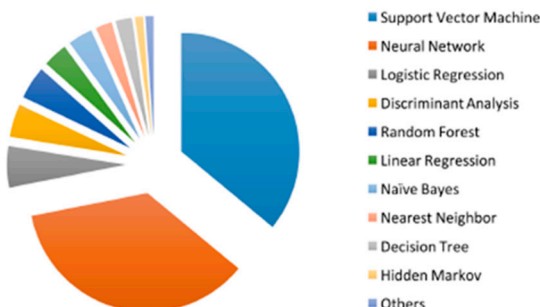

**Figure 5.** Machine learning algorithms used in healthcare sector [3].

### 5.1. Supervised Learning

In supervised machine learning, developers provide the algorithms with a prepared set of data as a training source. The task of the algorithms is only to recognize the pattern: Why does this information belong in category A and not in category B? Supervised learning is used for such algorithms that are intended to categorize natural data (photos, handwriting, language, etc.). In addition, so-called regression problems are a typical field of application for supervised learning. The algorithms should be able to make predictions, for example about the health of consumers, based on certain habits [32].

Let us assume that one would like to train algorithms to distinguish cancerous tumors from non-cancerous ones. The developers would then prepare a very large data set for this. This would contain scans that all already have a tag, i.e., belong to a category. You could imagine three different groups: cancerous, non-cancerous and other. It is important that the data collection also shows the greatest possible variance. Simply put, if you only have scans of non-cancerous tumors in your training set, the algorithm will assume that all tumors are cancerous. The data set should therefore map the actual range of variations as well as possible.

During training, the algorithm first receives the content (unsorted), decides independently and then compares it with the output specified by the developers. The system checks its own result against the correct one and draws conclusions from this that affect the next assessments during the training. The training continues until the machine and its assessments have come close enough to the correct results.

Which teaching method one should choose depends heavily on the later tasks of the algorithms. For categorization and regression problems, supervised learning is preferable to the other methods. In general, you can use monitored learning to train algorithms so that they are perfectly prepared for the area of application. Since you have complete control over the training material, you only need enough input and time to set the algorithms correctly. The emphasis here is clearly on input: the compilation must be large-scale. Since each element has to be provided with a label in supervised learning, this entails considerable effort for developers and scientists. The effort is relatively high, but it is also relatively easy to understand what is going on. While in unsupervised learning a lot remains unclear because the algorithms work for themselves without any real instructions, in supervised learning what the machine does is precisely defined. However, that can also be a disadvantage: the learned algorithms then also work within the restrictions that have been imposed on them. One cannot expect creative solutions.

### 5.2. Unsupervised Learning

Put simply, this learning method consists of an artificial neural network that analyzes a large amount of information and uses it to determine contexts, patterns and similarities between data. This process is based on different procedures. One of the techniques used in this type of learning is group analysis or clustering. In this case, the algorithms are responsible for forming groups autonomously in order to finally assign them to the data. For example, if the data is photos of cancerous and non-cancerous tumors, then the program sorts all photos of cancerous tumors into one category and those of noncancerous ones into another during unsupervised study. However, unlike supervised learning, this classification is not specified in advance. In unsupervised learning, algorithms make these decisions independently, based on the similarities and differences between the photos. Another method is mapping: in this case, the system combines the data for sorting based on the attributes they share. In this way, the job of algorithms is to find the relationships between the objects without requiring any resemblance between them [33–38].

Machine learning is not only applied to technological development, it also helps to make many areas of our daily routine easier and easier and to enrich daily life, business and research. In contrast to the other two learning methods (monitored and reinforced), developers are not involved in the training itself, which, in addition to possible time savings, has another advantage: unsupervised learning enables the recognition of patterns that no one could perceive before. Based on unsupervised machine learning, algorithms can therefore also develop creative ideas.

### 5.3. Reinforcement Learning

In contrast to the other two methods, reinforcement learning does not require any data beforehand. Instead, they are generated and labeled in a simulation environment in many runs in a trial-and-error process during the training. As a result, through reinforcement learning, a form of artificial intelligence is possible that can solve complex control problems without prior human knowledge [39–41]. Compared to conventional engineering, such tasks can be solved many times faster, more efficiently and, ideally, even optimally. Leading AI researchers refer to RL as a promising method for achieving artificial intelligence. Reinforced learning is essentially about learning through interactions with an environment. The key to solving reinforcement tasks is to find optimal guideline or value functions. The representation of a policy and the reinforcement learning method to be used depends specifically on the problem to be solved.

Reinforcement learning is a whole series of individual methods in which a software agent independently learns a strategy, working of reinforcement learning can be seen in Figure 6. The goal of the learning process is to maximize the number of rewards within a simulation environment. During training, the agent executes actions within this environment at every time step and receives feedback in each case. The software agent is not shown in advance which action is best in which situation. Rather, he receives a reward

at certain times. During the training, the agent learns in this way to assess the consequences of actions for situations in the simulation environment. Based on this, he can develop a long-term strategy to maximize the reward [32].

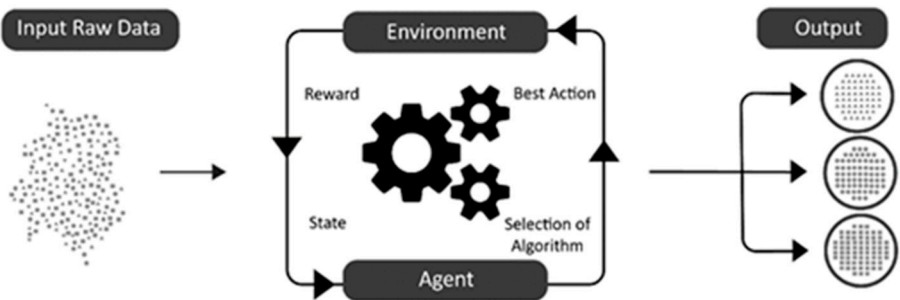

**Figure 6.** How reinforcement learning works.

Deep learning refers to machine learning with large artificial neural networks. These consist of several layers—typically input and output layers as well as some "hidden" layers in between (Figure 7). The individual layers in turn consist of a large number of artificial neurons that are numerically weighted, connected to each other and react to input from neurons from the previous layer. This weighting can be adjusted during the training process so that ever more accurate results are achieved [32]. If in the first shift a pattern is recognized, a pattern from the pattern is created in the second layer and recognized. This principle is continued in the next layers. The more complex the network (measured by the number of layers and connections between neurons and the neurons per layer), the more complex problems can theoretically be processed simultaneously. Deep learning has and is making remarkable breakthroughs in many areas, for example, it is used in the processing of natural language or in the recognition of objects. Data for machine learning bases diagnostics is sourced from different sources like diagnostic imaging, electrodiagnosis, genetic diagnosis, clinical laboratory, mass screening and others as shown in Figure 8.

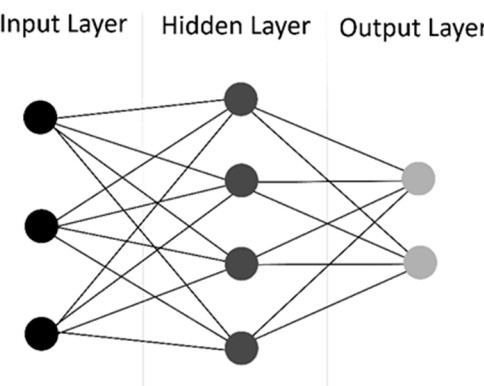

**Figure 7.** Simplified structure of artificial neural networks.

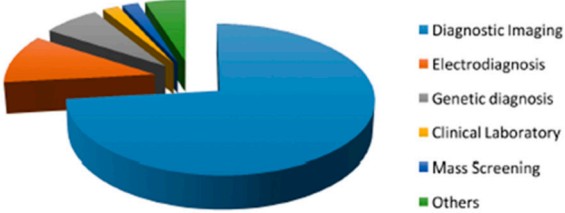

**Figure 8.** Data sources used for machine learning [3].

### 5.4. Machine Learning Applications in Smart Cities

Different machine learning algorithms can be applied across smart cities to dramatically optimize and improve application performance and yield improved results. A comprehensive Table 2 has been formulated below depicting possible smart cities use cases and the corresponding ML algorithm that can benefit the application.

**Table 2.** Machine learning in context of smart cities applications.

| Sr. | Smart Cities Application | ML Algorithm Type |
|---|---|---|
| 1 | Smart healthcare | Rule based |
| | Public safety | Pattern recognition |
| | Smart transportation | Semantic reasoning |
| 2 | Smart home | Multiagent learning |
| | Realtime traffic routing | Reinforcement learning |
| 3 | Smart pipelines | HMM |
| 4 | Energy | Semi-supervised deep reinforcement learning |
| | Water | |
| | Agriculture | |
| | Combatting pollution | |

### 5.5. Applications of Machine Learning and AI in Healthcare

More and more patients are recording their health data themselves via mobile apps and wearables. Using this data and processing it via an AI algorithm can prove to be very beneficial. With the help of data science, scientists and practicing physicians are developing new prevention approaches and researching rare hereditary diseases and previously unknown medical conditions. New diagnostic methods for personalized therapy are usually not possible without complex data analyses. Medical procedures for diagnostic imaging are already benefiting today from AI algorithms: these algorithms can be used by medical experts and support experts in making the images from radiographs, nuclear medicine procedures, magnetic resonance tomographies or ultrasound of organ systems (brain, lungs, skin, fundus, etc.) even more precise, faster and easier to more reliably analyze. For patients, AI-based applications can go hand in hand with more autonomy. Wearables allow them to set their own health values and monitor and use this as a basis to make their everyday life healthier. With direct access to personal data, the person has an additional information base in order to evaluate therapy options or possibly to make an initial self-diagnosis. In the long term, AI promises to efficiently evaluate large amounts of data and create new ones generating knowledge—for example in epidemiology, i.e., the investigation of connections and distributions of diseases and risk factors in the population. There are also new opportunities for the early detection of diseases by examining the genetic make-up (genome), the external appearance of an organism (phenotype), the proteins (proteome) or microorganisms (microbiome). In addition, vital parameters such as blood sugar or blood pressure become more efficient to monitor and control [33].

AI algorithms do not provide a black and white decision but much more; they calculate the probability of occurrence based on a model representation of reality. However, even models that have been trained extensively can only depict reality within certain limits. Therefore, an AI algorithm cannot derive a decision to act independently. Rather, the result represents the need for medical and nursing staff to take the required action. The following examples illustrate the benefits of AI for healthy people as well as for different patient groups. The categories cannot be clearly delimited from one another. Sometimes, a patient can be assigned to several classifications; for example, a stroke is an acute illness, but it usually results from chronic symptoms. AI has the potential to detect diseases at an early stage and thus reducing adverse consequences for the patient. Machine learning could

provide new insights into health data. The AI learns connections and identifies patterns in the data. As soon as the database expands, it grows the data available for machine learning. Patients could use this to better assess their risk of later illnesses and, if necessary, change their health behavior.

### 5.5.1. Health Monitoring and Prognosis

AI systems are expected to become increasingly popular in preventive medical checkups to evaluate health data and draw attention to possible risks. This would make it possible to identify risk groups for individual diseases more quickly and to carry out targeted examinations and screening. Medical checkups could be a perfect application in this scenario. AI can provide support in the early detection of diseases in particular. In fact, at an early stage, it is often difficult to identify rare diseases recognized by subtle symptoms.

Recently, Scientists of the EU project I-PROGNOSIS have developed a smartphone app that is intended to enable early detection of Parkinson's disease. As part of the research study, they are currently collecting data from healthy and sick study participants: everyday functions, such as holding a smartphone, telephoning, and taking photos, are stored as data in a Cloud. With the help of machine learning methods, the behavior is investigated and the user is asked to consult a doctor in the event of abnormalities (I-PROGNOSIS 2019).

Learning systems could determine individual recommendations for a patient's lifestyle changes and support them in self-management. Wearables are expected to play an important role in this and raise new questions about human-technology interaction. It can provide ongoing risk analysis and help set goals for a healthier everyday life and create training plans. The local data collected from end devices could be used to train global models. You would then be able to generate recommendations for action on several levels (e.g., individually, regionally and globally). It is known as distributed machine learning, in which different computers share their training of artificial intelligence [31].

### 5.5.2. Treatment of the Acutely Ill

AI is also increasingly being used in the treatment of those who are acutely ill. In oncology, doctors can use imaging procedures, carcinomas, metastases or surrounding areas suspected of being cancerous and identify the anomalies faster. The underlying procedure is carried out by means of a technology based on deep learning and the system marks suspicious areas in the image data. Doctors do not need to evaluate the image data manually and thus gain valuable time. AI can also help to improve the expressiveness of the images. Today, the high potential of a supporting AI in cancer diagnostics is emerging at a fast pace. The project "AI in Pathology" (November 2018 to October 2020) involves AI that supports diagnosis and colon cancer therapy. A support system analyzes tissue samples from colonoscopies. It identifies abnormalities, assesses the possible course of the disease and supplements if required and provides additional digital analysis information (BMBF 2018). In a 2019 published study, 157 dermatologists from twelve university clinics in Germany competed against a computer to detect skin tumors. The computer diagnosed 136 cases more accurately than humans [31].

### 5.5.3. Decision Support Systems

Decision support systems are being gradually used in hospitals and doctors' offices. In the future, they will help increase the success rate of various treatment options, while the doctors retain the authority to make decisions but expand their knowledge based on health data, study and research databases and examinations. AI-based applications point out new, relevant therapy options more quickly. The decision support system also delivers a treatment proposal, a justification that is also applicable to laypeople such as patients and that relatives can easily understand. This way, the treating doctors will be clearly relieved of the burden of acquiring knowledge and can devote more time to communicating with the patient. Even in emergency situations where an early diagnosis for health harm reduction

is crucial, AI could be helpful in the future. An AI-supported board computer with the help of a decision support system could be used to guide the emergency physicians in initial analyses and therapy recommendations. The basis for this is primarily the stored health data of the patient and the examination results on site. Complex pre-existing illnesses that an emergency doctor may not recognize under time pressure are dealt with by the data comparison more quickly. The doctor can use the start therapy recommendations quickly with an appropriate treatment. When the patient arrives at the hospital, a smart computer has already prepared the operating room. AI can also assist during the operation; AI tools provide the OR team with reliable knowledge that is based on the latest clinical studies. A decision support system calculates the success rate of various measures during the operation and gives recommendations for action on this basis. If the doctors change the information situation due to new findings the system can learn too [31].

The precision of the procedure reduces the risk for patients. Surgical robots are already performing minimally invasive operations with AI support, which allows a tremor-free operation with the highest precision. In the future, the robots will be equipped with additional image and sensor evaluation techniques, and they can be further trained and improved. It will then be possible to carry out the sub-steps of an operation, for example in body regions with many vessels, with a greater degree of autonomy. However, important decisions are still left to be made by the field experts. Researchers from Boston have already successfully tested a robotic catheter that operates autonomously using AI in humans and animals. An optical touch sensor and image processing algorithms make it possible to determine the exact location in the body. Scientists have already demonstrated the application in heart valve repair on an animal model [33].

### 5.5.4. Treatment of Chronic Illnesses

The chronically ill often have to take medication for the rest of their lives. Intelligent systems can help with the dispensing of medication and the setting of the dosage, thereby minimizing stress and side effects. For example, in the case of diabetes, your insulin needs increase when you change your diet, take medication or receive medical treatment. That is why so-called closed-loop glucose systems are currently being researched to develop the autonomous systems that take over the function of the pancreas. In such an intelligent system, the algorithm continuously accesses the data from a sugar-measuring device and controls an insulin pump on this basis so that the blood sugar control can be continuously adjusted

A chronic physical illness often leads to a psychological one hence increasing the burden for the sick person himself or his relatives, and pure mental illnesses are also often chronic. AI has the potential to diagnose psychological problems at an early stage and to support treatment. It can provide information on the basis of which the affected person, relatives, nursing staff or doctors can provide healing or at least take soothing measures. MIT scientists have developed a model that uses an artificial neural network that can recognize depressive changes based on speech patterns. The model was made with data from 142 clinical interviews. This would make one possible application on the smartphone; the text and voice of the user is analyzed for abnormal patterns and if there are any signs of abnormal behavior [30].

### 5.5.5. Respite Care

Many old people's homes are now documenting digitally, and outpatient care services are also using the technology more and more frequently. At the same time, nurses are increasingly interested in the new technologies. The acquisition and maintenance costs are currently still very high. In addition, caring for people is fundamentally a very complex process that cannot easily be taken over by robots, because human affection and empathy cannot be totally replaced by robots. Although the care sector is even less digitized than other sectors in the health sector, there are already promising applications for artificial intelligence. For example, it is conceivable that AI-supported speech recognition could

support the care documentation. This could further simplify the time-consuming task. The latest research results from the field of AI-supported rehabilitation robots, which are to be used to improve motor skills after neurological diseases, are also very promising. Based on individual data, learning processes can create an optimal and adaptable training program for the patient [33].

In RECUPERA, a rehabilitation project, the Robotics Innovation Center of the German Research Center for Artificial Intelligence (DFKI) achieved a breakthrough in the area of rehabilitation robotics. Together with Rehaworks GmbH, the project members developed a mobile exoskeleton for upper body assistance, which is specially designed for rehabilitative therapy after a stroke (DFKI 2018). By measuring bio-signals (e.g., brain and muscle activities or the detection of viewing directions) in combination with contextual factors, AI-supported robots could support the rehabilitation of stroke patients in the future. When the motor skills are disturbed after a stroke, these systems recognize movement intentions and set them around. For example, the patient may no longer be able to cope with raising his right arm—the AI evaluates the brain activity and can use it to determine the problem and implement the solution using robotics. In this way you can achieve rehabilitation success: stroke patients can improve and regain their motor skills faster. Such rehabilitation robots show the high performance of learner systems, because large amounts of data have to be processed efficiently in the shortest possible time with extremely low energy consumption so that the rehabilitation robot can be controlled on the basis of the bio-signals.

## 6. Conclusions

In the bigger picture this paper explains the use of AI, blockchain, machine learning and IoT technologies for the development of smart healthcare systems and broadly for the development of smart cities that can truly enhance and optimize the accuracy of expected results. Sensor networks, IoT and machine learning therefore offer untapped potential to relieve doctors and better identify diseases. It is precisely because the health systems are facing major challenges due to an aging society, financial bottlenecks and growing mountains of (analog and digital) data, the use of modern technologies is becoming increasingly inevitable. This will possibly lead to the increased use of robots in everyday medical practice. In the near future, even items of personal health use such as toothbrushes can be equipped with IoT-enabled sensors that will eventually lead to an improved quality of life in general and even help save countless lives.

The future of healthcare is very promising, considering the rapid development in sensor technology, AI and machine learning. For patients, hospitals and doctors as well as for medical device manufacturers, there are not only new opportunities but even the obligation to make use of the Internet of Things. It is obvious that challenges and substantial risks have to be mastered. Throughout the reviewed literature there is a consistency for use of smart technologies in smart cities and in particular healthcare, and AI and blockchain technologies are key driving factors for enhancement and improvement of overall user experience of smart cities. Although there are potential downsides of artificial intelligence and machine learning technologies in the context of smart cities, they still also have a potential to change the way we know smart healthcare and smart cities as of now.

**Funding:** This research received no external funding.

**Data Availability Statement:** Not Applicable.

**Conflicts of Interest:** The authors declare no conflict of interest.

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
