# Peer review of "IoT for Smart Cities: Machine Learning Approaches in Smart Healthcare—A Review"

_futureinternet, doi:10.3390/fi13080218_

Round 1

Reviewer 1 Report

This is an interesting paper, but there is a major lacuna in the literature review and theoretical framework.

First, there is a problematic assumption of the benefits of smart tech and IoT. This means that the authors are not acknowledging all the important critical literature on smart cities. These are some key references:

Cugurullo, F. (2018). Exposing smart cities and eco-cities: Frankenstein urbanism and the sustainability challenges of the experimental city. Environment and Planning A: Economy and Space, 50(1), 73-92.

Kitchin, R., & Dodge, M. (2019). The (in) security of smart cities: Vulnerabilities, risks, mitigation, and prevention. Journal of Urban Technology, 26(2), 47-65.

Karvonen et al. (2018). Inside smart cities: Place, politics and urban innovation. Routledge.

Second, there is again a problematic and uncritical understanding of AI and of how AIs and machines learn. Key references that would help fill this lacuna are:

Cugurullo, F. (2021). Frankenstein Urbanism: Eco, Smart and Autonomous Cities, Artificial Intelligence and the End of the City. Routledge.

Also the work of Yigitcanlar on the sustainability of artificial intelligence: An urbanistic viewpoint from the lens of smart and sustainable cities, is an important and useful one.

Author Response

First of all, we would like to thank the reviewer for the kind remarks and suggestions. After incorporating these suggestions, they have clearly made our manuscripts more polished in a better way. We appreciate the recommendations.

Here is the detail of all the suggestions and how we incorporated them in our manuscript, a point-to-point response to the comments has been uploaded. The updated manuscript with ‘Track Changes’ on is ready for submission as the editor allow, so reviewer can also view from the original file.

Reviewer 2 Report

The authors studied the review on the Internet of Things for Smart Cities. Manuscript written work needs improvement.

  1. In the Abstract, It is not clear that what is your role or contribution. Redraft the abstract part.
  2. In section 1, add more motivational parts and contributions in pointable form.
  3. All figures in the manuscript are blure, so redraw the all figures.
  4. Section 2 [Related work], provide a comparison table between existing studies with limitations, and solutions with proper methods.
  5. The summarization paragraph adds in section 2 also.
  6. What is the use of Section 3 [Wireless sensor networks]? It seems like theoretical or bookish form, you can make some useful form such as WSNs for Smart City or WSNs for IoT applications. And add some subsections in these sections. Also, add Taxonomy Diagrams.
  7. Line number 228, What is this? It is a paragraph or something. I am not understanding.
  8. If you are taking figures from the Internet, then you redraw again compulsory. It is a good habit in research.
  9. In Section 5, you can use ML for Smart City or ML for IoT Applications and add some taxonomies or tables.
  10. Carefully check your manuscript organization and improve it.
  11. Check Line no. 602, this paragraph does not end. Why? A lot of grammatical mistakes in the manuscript. Show the manuscript to the English Checking company. Extensive editing of English language and style required.
  12. In the Conclusion part, the Future scope adds at the end of the conclusion instead of starting.
  13. Cross-reference all citations and ensure that they match accordingly. Recent reference must be added, the following is recommended:
  • Mitigating and monitoring smart city using internet of things.
  • Machine learning-based distributed big data analysis framework for next-generation web in IoT.
  • DeepBlockScheme: A Deep Learning-Based Blockchain Driven Scheme for Secure Smart City.
  • Energy efficiency in internet of things: an overview.
  • OTS Scheme Based Secure Architecture for Energy-Efficient IoT in Edge Infrastructure.

Author Response

(The authors gave the same response as above.)

Reviewer 3 Report

This article has a misleading title and according to the main text it presents a base line (review) study for the healthcare sector and innovative computing and network technologies.

I suggest to authors to improve their article in order to present the content more clearly, to be more readable for the reader, and to be accepted for publication. My comments are as follows:

  1. Please rewrite the abstract. Only the last 3 lines has a meaning for me (in relation with the title and the main text)
  2. Make improvements in section 1 and 2, maybe combine them. After the abstract the reader reads about many different things which do not have a normal order. Also in case that you present a review paper please describe your methodology, databases searched and keywords.
  3. None of the Figures do not have references in the text. So it's not clear which is their purpose in your work.
  4. Please make it more clear why you reference articles [13-17] in your work.
  5. Make right use of AI and ML. ML is a subset of AI.
  6. Also, the text has many typos.
  7. Finally, the authors present technologies and applications but they do not discuss their findings. Please add a discussion section or improve your Conclusions.

Author Response

(The authors gave the same response as above.)

Round 2

Reviewer 2 Report

According to the previous version manuscript, the authors improved and revised the latest version, but this version also has some minor revisions:

  1. Carefully check figures, now the version file has also blure figures, specific Fig. No. 2.
  2. Some figures are left alignment, and some are middle alignment. Why?
  3. 2 name should be “Graphical Representation of WSNs for Smart Cities”. Check other Figures' naming also.
  4. Check Line No. 429, and 432 texts.
  5. Check Line No. 126 and 130. What is [5] identified? It is wrong, you can use the example Salim et al [5] identified. Please check the whole manuscript and improve this mistake.
  6. In Section 5, you can use ML for Smart City and add some taxonomies or tables, if possible.
  7. Check the previous comment No. 13. All papers must cite in the published version paper.

Author Response

First of all, we would like to thank the reviewer for the kind remarks and suggestions. After incorporating these suggestions, they have clearly made our manuscripts more polished in a better way. We appreciate the recommendations.
Here is the detail of all the suggestions and how we incorporated them in our manuscript, a point-to-point response to the comments has been uploaded. The updated manuscript with ‘Track Changes’ on is also submitted to the system, so reviewer can also view from the original file.

Reviewer 3 Report

The authors responded in almost all open issues. I agree to publish this work.

Author Response

First of all, we would like to thank the reviewer for the kind remarks and suggestions. After incorporating these suggestions, they have clearly made our manuscripts more polished in a better way. We appreciate the recommendations.
Here is the detail of all the suggestions and how we incorporated them in our manuscript, a point-to-point response to the comments has been uploaded. The updated manuscript with ‘Track Changes’ on is also submitted to the system, so reviewer can also view from the original file.

This manuscript is a resubmission of an earlier submission. The following is a list of the peer review reports and author responses from that submission.

Round 1

Reviewer 1 Report

The paper is aiming to provide a survey of AI, IoT and WSN approaches in smart city applications with a focus on healthcare. Authors also touch a few other topics such as blockchain but did not provide detailed discussion and how it fits in the big picture together with IoT and AI aspects. 

Although the paper is mainly geared towards healthcare, however the title does not match the content. Furthermore, the paper fails to provide a comprehensive survey of proposed topics and does not present a significant contribution to the field. A survey paper is expected to explore the current state of the art and offer deeper insight about current methods and technologies. Unfortunately, the explored literature is not enough and the paper includes a small number of references (including inconsistent reference formats).

My main concern is about the adopted content without reference. Some figures are adopted from other publications without providing the reference. For instance figures 1, 3, 5, and 6 are adopted from the following paper: 
Jiang, Fei, Yong Jiang, Hui Zhi, Yi Dong, Hao Li, Sufeng Ma, Yilong Wang, Qiang Dong, Haipeng Shen, and Yongjun Wang. "Artificial intelligence in healthcare: past, present and future." Stroke and vascular neurology 2, no. 4 (2017).

Reviewer 2 Report

This is a really interesting contribution that explores the increasing role that machine learning and AI are playing in smart-city initiatives.

The paper is well-written and I only have to main revisions to recommend:

First, the author should better define the key concepts such as AI and explain how these concepts are becoming part of smart urbanism (because this is a recent phenomenon that requires more background information for the reader). On this regard, I recommend these two papers:

Batty, M. (2018). Artificial intelligence and smart cities. Environment and Planning B, 45(1), 3-6.

Cugurullo, F. (2020). Urban artificial intelligence: From automation to autonomy in the smart city. Front. Sustain. Cities, 2(38), 1-14.

Second, the authors should be more critical in relation to the use of AI and machine learning in smart cities. AI tech has a lot of potential downsides that have been discussed in the literature on smart urbanism. On this regard I recommend in particular this book:

Cugurullo, F. (2021). Frankenstein Urbanism: Eco, Smart and Autonomous Cities, Artificial Intelligence and the End of the City. Routledge.

As well as the work of Yigitcanlar on the sustainability of artificial intelligence from the lens of smart and sustainable cities.

Reviewer 3 Report

Based on the manuscript organization, I will not recommend the further step because it has various issues:

  1. The main issue is that manuscript has not any novelty.
  2. The abstract is not clear according to the title (What is the main contribution?)
  3. Add contribution and organization in Section 1.
  4. Some figures are taken directly from the internet and no cite figures.
  5. What the role the Section 2 and 3? I did not understand.
  6. Where is the comparison of your work to the existing research?
  7. The manuscript has a lot of grammatical mistakes and English problems.
  8. Citations are not proper. Why you are using double-double citation like [1] [1], [10] [10],….
  9. Why you are using citation in the conclusion part.

I encourage the authors that the revised version manuscript can resubmit on the portal.